# The Interaction of miR-378i-Skp2 Regulates Cell Senescence in Diabetic Nephropathy

**DOI:** 10.3390/jcm7120468

**Published:** 2018-11-22

**Authors:** Yi-Chun Tsai, Po-Lin Kuo, Mei-Chuan Kuo, Wei-Wen Hung, Ling-Yu Wu, Wei-An Chang, Ping-Hsun Wu, Su-Chu Lee, Hung-Chun Chen, Ya-Ling Hsu

**Affiliations:** 1Graduate Institute of Clinical Medicine, College of Medicine, Kaohsiung Medical University, Kaohsiung 807, Taiwan; lidam65@yahoo.com.tw (Y.-C.T.); kuopolin@seed.net.tw (P.-L.K.); mechku@kmu.edu.tw (M.-C.K.); esther906@gmail.com (L.-Y.W.); 960215kmuh@gmail.com (W.-A.C.); 970392@mail.kmuh.org.tw (P.-H.W.); 2School of Medicine, College of Medicine, Kaohsiung Medical University, Kaohsiung 807, Taiwan; 3Faculty of Renal Care, College of Medicine, Kaohsiung Medical University, Kaohsiung 807, Taiwan; chenhc@kmu.edu.tw; 4Division of General Medicine, Kaohsiung Medical University Hospital, Kaohsiung 807, Taiwan; 5Division of Nephrology, Kaohsiung Medical University Hospital, Kaohsiung 807, Taiwan; suchle5910@gmail.com; 6Division of Endocrinology and Metabolism, Kaohsiung Medical University Hospital, Kaohsiung 807, Taiwan; hung4488@ms57.hinet.net; 7Division of Pulmonary and Critical Care Medicine, Kaohsiung Medical University Hospital, Kaohsiung 807, Taiwan; 8Graduate Institute of Medicine, College of Medicine, Kaohsiung Medical University, Kaohsiung 807, Taiwan

**Keywords:** cell senescence, miR-378i, Skp2, diabetic nephropathy, proximal tubular epithelial cell

## Abstract

Diabetic nephropathy (DN) is the major cause of end stage renal disease. Proximal tubular epithelial cell (PTEC) injury occurs early in diabetic kidney, and it is correlated with consequent renal failure. Cellular senescence participates in the pathophysiology of DN, but its role remains unclear. We conducted a cross-disciplinary study, including human, in vivo, and in vitro studies, to explore the novel molecular mechanisms of PTEC senescence in DN. We found that HG induced cell senescence in PTECs, supported by enhanced β-galactosidase staining, p53 and p27 expression, and reduced cyclin E levels. Transcriptome analysis of PTECs from a type 2 diabetic patient and a normal individual using next generation sequencing (NGS) and systematic bioinformatics analyses indicated that miR-378i and its downstream target S-phase kinase protein 2 (Skp2) contribute to HG-induced senescence in PTECs. High glucose (HG) elevated miR-378i expression in PTECs, and miR-378i transfection reduced Skp2 expression. Urinary miR-378i levels were elevated in both db/db mice and type 2 diabetic patients, whereas decreased Skp2 levels were shown in proximal tubule of db/db mice and human DN. Moreover, urinary miR-378i levels were positively correlated with urinary senescence-associated secretory phenotype cytokines and renal function in in vivo and human study. This study demonstrates that the interaction between miR-378i and Skp2 regulates PTEC senescence of DN. miR-378i has the potential to predict renal injury in DN. These findings suggest future applications in both therapy and in predicting renal dysfunction of DN.

## 1. Introduction

Diabetes mellitus (DM) is an increasingly serious threat to human health and it must be regarded as an important public health issue [1]. Diabetic nephropathy (DN) has been the major cause of end stage renal disease (ESRD) worldwide, and it creates a huge burden on health-care systems [2]. Proximal tubular epithelial cells (PTECs) are the major target exposed to injury in glucose-induced metabolic disorder [3]. Proximal tubular changes occur early in the diabetic kidney, and have the potential to facilitate early prediction of long-term outcomes [4].

Disease-induced stress or damage may trigger cell senescence, and senescent cells may then drive and accelerate disease progression [5]. The mechanism of cell senescence involves cyclin-dependent kinase (CDK) inhibition to prevent cells from replicating and passing on a damaged genome [4,6]. Tumor suppressor p53 and CDK inhibitors, including p16, p21, and p27, participate in prototypical senescent arrest or senescent-like growth arrest [7]. Additionally, senescent cells are highly active metabolically, whereby they develop a multi-component senescence-associated secretory phenotype (SASP), such as interleukin-6 (IL-6), IL-8, and monocyte chemoattractant protein-1 (MCP-1) [8,9], which contains transcriptional activation and numerous secretory factors that induce biological activities in neighboring cells and surrounding tissue [10]. SASP may explain why senescent cells can initiate a deleterious positive feedback mechanism by promoting the spread of senescence to nearby cells [11]. Hyperglycemia has been reported to induce SASP activation [12].

Accumulating evidence indicates that cellular senescence might play a role in the pathophysiology of type 2 DM [12,13,14]. Senescent cells cannot turnover and promote tissue repair, and may instead contribute to microvascular complications that were observed in type 2 DM [14]. A previous study has found senescence-associated β-galactosidase (SAβ-Gal) activity in mesangial cells, podocytes, and renal tubule cells in type 2 human diabetic kidney [15]. Hyperglycemia causes cellular senescence via sodium/glucose cotransporter 2 (SGLT2)- and p21-dependent pathway in proximal tubules in the early stage of DN [16]. These findings suggest that cellular senescence may participate in the pathogenesis of DN [5].

microRNAs (miRNAs) are endogenously encoded, conserved small RNAs that regulate gene expression by inhibiting the protein translation of target mRNAs by binding to the 3’-untranslated region (3’UTR) of mRNA [17]. Some miRNAs, such as miR-125b, miR-126, miR-21, miR-22, miR-29, miR-210, miR-34a, miR-449a, miR-494, and the miR-200 family, have been found to play a role in the pathogenesis of cellular senescence [18,19]. Regretfully, there have been no reports on the regulation of miRNAs in cell senescence of DN until now.

In this study, we utilized RNA-seq and bioinformatics to determine the novel signaling involved in cell senescence—miR-378i-S-phase kinase protein 2 (Skp2) pathway in DN. Next, we conducted a cross-disciplinary study to investigate whether miR-378i-Skp2 signaling participates in PTEC senescence in DN. Our study provides new insight to understanding the unique crosstalk accounting for PTEC dysfunction owing to cell senescence by HG in DN.

## 2. Materials and Methods

### 2.1. Cell Lines and Cell Cultures

Primary proximal tubular epithelial cells (PTECs) of a DM patient and a normal individual (Lonza Walkersville Inc., Walkersville, MD, USA) were cultured in Clonetics^™^ REGM^™^ BulletKit^™^ (CC-3190). Human PTECs (ATCC PCS-400-010) were cultured in Renal Epithelial Cell Basal Medium (ATCC PCS400030^™^) plus 0.5% Fetal Bovine Serum (FBS), according to the manufacturer’s suggestion. Cells were treated with normal glucose (NG, 6.2 mM) and HG (30 mM) for the indicated times. Human embryonic kidney (HEK) 293 cells (ATCC^®^ CRL-1573^™^) were purchased from the American Type Culture Collection (Manassas, VA, USA) and were cultured in DMEM containing 10% FBS.

### 2.2. Senescence-Associated β-Galactosidase (SAβ-Gal) Staining

The cellular senescence characteristics of human PTECs were identified by serial observations of their morphological changes using light microscopy (Nikon ECLIPSE TE20000-S, Nikon, Tokyo, Japan). Additionally, SAβ-Gal activity of PTECs that had undergone NG and HG treatment for seven and nine days was measured using a senescence cell histochemical staining kit (Sigma #cs0030-1kt, St. Louis, MO, USA), according to the manufacturer’s protocol.

### 2.3. Western Blot Analysis

PTECs were lysed in RIPA Lysis buffer (EMD Millipore, Burlington, MA, USA). Equal amounts of protein were subjected to 9–11% SDS-PAGE for electrophoresis, followed by transfer onto a PVDF membrane. The membrane was blocked with 5% blocking buffer and sequentially immunoblotted with each primary antibody overnight at 4 °C. The membrane was washed with TBS/Tween-20 (0.2%) and then incubated with a horseradish peroxidase (HRP)-conjugated secondary antibody. The corresponding bands were detected using a chemiluminescent HRP substrate kit (EMD Millipore, Burlington, MA, USA). The chemiluminescence signal was analyzed using Proteinsimple + Fluorchem Q (Alpha Innotech, San Leandro, CA, USA). The densitometry of the bands was analyzed using Image J software version 1.52. p53 Antibody (Catalog #9282), p27 Antibody (Catalog #3686s), Skp2 Antibody (Catalog #4358), Cyclin E1 (HE12) Mouse mAb (Catalog #4129), and Cyclin E2 Antibody (Catalog #4132) were obtained from Cell Signaling (Danvers, MA, USA). GAPDH antibody (Catalog #MAB374) was obtained from EMD Millipore (Burlington, MA, USA).

### 2.4. RNA Sequencing

RNA samples from human primary PTECs and normal primary PTECs were collected and profiled by NGS. Total RNA from harvested cells was extracted by Trizol^®^ Reagent (Invitrogen, Carlsbad, CA, USA), according to the manufacturer’s instructions. The quality of extracted RNA was analyzed via OD260 detection using an ND-1000 spectrophotometer (Nanodrop Technology, Wilmington, DE, USA) and samples were then readied for further RNA preparation and sequencing analysis of small RNA-seq by Welgene Biotechnology Company (Welgene, Taipei, Taiwan). The quality of extracted RNA was confirmed by RNA integrity number (RIN) using Agilent Bioanalyzer (Agilent Technology, Santa Clara, CA, USA). To construct the small RNA library and perform deep sequencing, samples were prepared using an Illumina sample preparation kit according to the TruSeq Small RNA Sample Preparation Guide. Total RNA was ligated with 3′ and 5′ adaptors and was reverse-transcribed into cDNA by polymerase chain reaction (PCR) amplification. The harvested cDNA constructs were fractionated by size on a 6% polyacrylamide gel electrophoresis, and the bands containing 18–40 nucleotide RNA fragments (140–155 nucleotide in length with both adapters) were purified. Libraries were then sequenced on an Illumina GAIIx instrument (75-cycle single read) and the sequencing results processed with Illumina software BCL2FASTQ version 2.20. The differentially expressed mRNAs between cells were set at fold change >2.0, fragments per kilobase of transcript per million (FPKM) >0.3 for mRNA and reads per million (RPM) >10.0 for miRNA. 

### 2.5. Database for Annotation, Visualization and Integrated Discovery (DAVID) Bioinformatics Resources

The updated DAVID Bioinformatics Resources (https://david.ncifcrf.gov/) is a public resource that merges numerous public bioinformatics resources and it provides many tools to analyze large gene lists from genomic studies. After a gene list is uploaded, researchers can gain gene-term enrichment analysis and an overall notion of the biological functions associated with the gene list of interest [20,21].

### 2.6. Search Tool for the Retrieval of Interacting Genes (STRING)

The STRING database (version 10.5) (https://string-db.org/) includes over 9.6 million proteins, 2000 organisms, and 1380 million interactions that offer the analysis and integration of direct and indirect protein-protein interactions (PPI) and their functional associations [20]. Differentially expressed genes are uploaded, and those interactions with at least medium confidence (interaction score >0.4) were selected [22].

### 2.7. Ingenuity Pathway Analysis (IPA)

The IPA software (IPA, Ingenuity systems, Redwood City, CA, USA) provides “Core Analysis” for genes/proteins. Network analysis and canonical pathways that were obtained from Core Analysis results offer the interaction of genes/proteins. Network graphics with further overlay analysis can be generated by the software [20,23]. The IPA software also offers miRNA-mRNA filter analysis to match miRNAs and their potential targets of mRNAs according to the repression strength.

### 2.8. TargetScan and MiRmap Database

The TargetScan version 7.2 (http://www.targetscan.org) and miRmap (https://mirmap.ezlab.org) software rank potential targets of a specific miRNA based on Context++ score percentiles and miRmap scores, which indicates the repression strength of a miRNA target [20,24]. The TargetScan and miRmap web also can provide miRNA target predictions for different organisms.

### 2.9. RNA Extraction and Reverse Transcription PCR (RT-PCR)

Urine Exosome RNA Isolation Kit (Catalog #47200, Norgen, Thorold, ON, Canada) was used to isolate exosomal RNAs of human and mice urine (5 mL), following the manufacturer’s protocol. Oligo (dT) primer and reverse transcriptase (RT; Takara, Shiga, Japan) were utilized to prepare the cDNA after RNA extraction. miRNAs were reverse transcribed using the Mir-X^™^ miRNA First Strand Synthesis Kit (Catalog #638313 Takara, Shiga, Japan). We used SYBR Green on the StepOnePlus Real time PCR system (Applied Biosystems, Foster City, CA, USA) to analyze quantitative RNA and miRNA. PCR reaction was carried out with the following temperature profile (95 °C for 10 min, followed by 40 cycles at 95 °C for 15 s and 60 °C for 1 min). Relative expression levels of the mRNA and miRNA in cells were normalized to GAPDH or U6, respectively. miRNAs in exosomes that were isolated from urine were normalized with cel-miR-39 (Exiqon, Vedbaek, Denmark) as a spike-in control, and were then compared with a reference sample. Relative expression was presented using the 2^−ΔΔCt^ method. The primers used are listed in Table 1.

### 2.10. Transient Transfection

miR-378i mimic (100 nM), and miR-negative control of mimic (miR-NC, 100 nM) (Dharmacon, Lafayette, CO, USA) were transfected into cells by using DharmaFECT No. 1 Transfection Reagent (Catalog #T-2001-03, Dharmacon, Lafayette, CO, USA), following the manufacturer’s instructions.

### 2.11. Experimental Animals

Five-week-old male C57BL/6 mice (*n* = 3), pathogen-free male db/m mice (non-diabetic animal model) (*n* = 6), and db/db mice (type 2 DM animal model) (*n* = 6) were purchased from the National Laboratory Animal Center in Taiwan, and were reared at the experimental animal center of Kaohsiung Medical University, under controlled the temperature at 25 ± 5 °C and the humidity of 55 ± 5% in regular light/dark cycles. At the 12th week, the mean body weight of C57BL/6 mice, db/m mice, and db/db mice was 23.3 ± 0.6 g, 27.8 ± 0.7 g, and 40.4 ± 4.7 g, respectively. Blood and urine samples were collected and kidneys harvested at the 12th week. The kidneys were fixed in 4% paraformaldehyde. All animal experiments in this study were in strict agreement with Kaohsiung Medical University and the Use Committee.

### 2.12. Human Study Participants

One hundred and seven type 2 DM patients with estimated glomerular rate (eGFR) ≥ 30 mL/min/1.73m^2^, and 45 healthy volunteers were enrolled. Study participants were asked to fast for at least 12 h before the collection of urine and blood samples. All urine and blood samples were aliquoted and stored in a −80 °C freezer.

Diabetes was defined as a medical history of diabetes or the use of anti-diabetes agents. Demographic and medical data, including age, gender, and angiotensin-converting enzyme (ACEI)/angiotensin II receptor blocker (ARB) usage, which has the effect of reducing albuminuria, were obtained from medical records and interviews with study participants. Kidney tissue samples were collected from four DN patients scheduled for kidney biopsy and four patients receiving nephrectomy because of upper tract urothelial carcinoma. The study was approved by the Institutional Review Board of the Kaohsiung Medical University Hospital (KMUHIRB-G(I)20160036, KMUHIRB-G(I)20170037, KMUHIRB-20130089).

### 2.13. Laboratory Data Measurement and Quantification of Urinary IL-6, IL-8, MCP-1 and Urinary Albumin/Creatinine Ratio (ACR) in Mice and Humans

Serum creatinine was measured by the compensated Jaffé (kinetic alkaline picrate) method in a Roche/Integra 400 Analyzer (Roche Diagnostics, Mannheim, Germany) using a calibrator traceable to isotope-dilution mass spectrometry [25]. The value of eGFR was calculated using the four-variable equation in the Modification of Diet in Renal Disease (MDRD) study [26]. Serum blood urea nitrogen (BUN) was measured using a glutamate dehydrogenase and urease kinetic method. Glucose was analyzed by six-carbon glucokinase enzyme method.

Levels of IL-6 and MCP-1 in the urine of mice and humans, and IL-8 in the urine of humans, were measured using Magnetic Luminex^®^ Assay (human Catalog #FCST03 and mice Catalog #LXSAMSM-02 from R&D System, Minneapolis, MS, USA). Levels of urinary albumin of humans and mice were measured using immunoturbidimetric assay with Tina-quant Albumin Gen.2 (ALBT2, Roche Diagnostics, Indianapolis, IN, USA). Concentrations of urine creatinine (Cr) in humans and mice were measured using the enzymatic method (creatinine plus version 2, CREP2, Roche Diagnostics, Indianapolis, IN, USA). Concentrations of IL-6, IL-8, and MCP-1 were corrected by urine Cr before statistical analysis. Normoalbuminuria was defined as urinary ACR <30 mg/g; microalbuminuria was defined as urinary ACR ≥30 mg/g and <300 mg/g; and, macroalbuminuria was defined as urinary ACR ≥300 mg/g. DN was defined as diabetic patients with urinary ACR ≥30 mg/g.

### 2.14. Immunohistochemistry Stain of Humans and Mice Kidneys

Kidneys were fixed in 4% paraformaldehyde for the detection of renal morphology and immunohistochemical (IHC) staining. p53 antibody (1:200, Catalog #Ab31333, Abcam, Cambridge, UK) and Skp2 antibody (1:100, Catalog #15010-1-AP, Proteintech, Rosemont, IL, USA) were stained, respectively. Stained kidneys were observed using Leica (ICC50 HD, Buffalo Grove, IL, USA). The images quantification was performed using the IHC Profiler Plugin of ImageJ Software (https://imagej.nih.gov/ij/) [27].

### 2.15. Statistical Analysis

The categorical variables were expressed as percentages. Differences in the distribution of categorical variables were tested using the Chi-square test. The continuous variables were expressed as mean ± S.E.M or median (25th, 75th percentile), as appropriate. The significance of differences in continuous variables between the groups was tested using Student’s *t*-test or one-way analysis of variance (ANOVA), followed by the post hoc test adjusted with a Tukey correction, as appropriate. The association among continuous variables was examined by Spearman correlation. 

## 3. Results

### 3.1. HG Induces PTEC Senescence in DN

Cell senescence has been reported to participate in the pathophysiologic mechanism of DN [5]. To investigate cell senescence in human PTECs in DN, the cells were treated with NG and HG for seven and nine days, and SAβ-Gal staining was used to examine cell senescence. We found that cell morphologies changed from thin and spindle-shaped to flattened, irregular shapes with increased intracellular debris after HG treatment compared to those after NG treatment. The number of cells with SAβ-Gal staining was also elevated under the HG condition when compared to the NG condition (Figure 1A). 

Western blot was used to evaluate cell senescence markers, such as p53, p27, and cell cycle-related factor cyclin E in human PTECs under NG or HG conditions. HG elevated p53 and p27 expression and reduced cyclin E1 and cyclin E2 (Figure 1B–E) expression in PTECs after treatment for three and four days. We further investigated whether cell senescence occurred in mice and humans. IHC results revealed that p53 expression (high positive and positive proportion) in the proximal tubular area was higher in diabetic db/db mice when compared to normal C57B6 mice and non-diabetic db/m mice (Figure 1F). Also, elevated p53 expression was found at the proximal tubular area of human DN when compared to normal individuals (Figure 1G). Therefore, we suggest that PTEC senescence may contribute to the development of DN.

### 3.2. Identification of Differentially Expressed Genes Associated with Cell Senescence in PTECs of DM Patients and Normal Individuals

To investigate genes that are potentially associated with PTEC senescence in DN, RNA samples from PTECs of a normal individual and a DM patient were collected and RNAs were profiled by next generation sequence (NGS) and candidate regulators of PTEC senescence were detected by bioinformatics (Figure 2A). The differentially expressed genes in normal and diabetic PTECs are displayed as a volcano plot in Figure 2B. Differentially expressed protein-coding genes were screened with >2.0-fold-change between normal and diabetic PTECs, with a threshold setting of >0.3 FPKM. After screening, 612 mRNAs with significant <2.0-fold-change of PTECs between diabetic and normal PTECs were identified. Of 612 mRNAs, 280 mRNAs had up-expression and 332 mRNAs had down-expression in diabetic PTECs.

To understand the biological functions of the 612 differentially expressed genes in diabetic PTECs, these genes were first input into the DAVID database for enrichment analysis, using the Kyoto Encyclopedia of Genes and Genomes (KEGG) pathway. The top seven enriched terms of KEGG pathway are shown in Figure 2C. Cell cycle was the most greatly enriched biological processes, and p53 pathway was also one of the top seven of KEGG pathway. These results support the hypothesis that HG induces human PTEC senescence in vitro.

We further utilized the STRING database to identify the potential interaction among these differentially expressed genes associated with cell cycle regulation, based on the results from the DAVID database (Figure 2D). Of the 14 genes related to cell cycle, Skp2 had the highest correlation with various cell cycle regulators, such as cyclin D2 (CCND2), cyclin B1 (CCNB1), and cyclin-dependent kinase inhibitor 1C (CDKN1C), as determined by the edge confidence of STRING database. We also used Core Analysis of IPA to analyze 612 differentially expressed genes in diabetic PTECs, with five networks that are related to cell development and cell cycle (Table 2). Skp2 had played a role in the regulation of cell cycle through the upregulation of cyclins (Figure 2E). Therefore, the loss of Skp2 may cause cell cycle arrest, one factor of the senescence phenomenon in human PTEC.

### 3.3. Decreased Skp2 Expression Contributes to Cell Senescence in DN

According to the results of our bioinformatic analysis, Skp2 participated in the regulation of the cell cycle of diabetic PTECs. We examined whether HG affects Skp2 expression in DN from in vitro, in vivo, to human studies. Decreased Skp2 mRNA was seen in diabetic PTECs compared to normal PTECs (Figure 3A). Skp2 mRNA levels decreased in human PTECs that were treated with HG for three days (Figure 3B). Western blot analysis showed that HG reduced Skp2 protein expression in human PTECs on days 3 and 4 (Figure 3C). Furthermore, the IHC stain was used to investigate Skp2 expression in mice and humans, and it revealed that the Skp2 levels were lower in the proximal tubule area of db/db mice as compared to that of C57B6 mice and db/m mice (Figure 3D). Consistently decreased levels of Skp2 were observed at the proximal tubular area of human DN patients than that of normal human kidney (Figure 3E). Since the physiological function of Skp2 assists the cell cycle and HG diminishes Skp2 expression in PTECs, we inferred that decreased levels of Skp2 might cause PTEC senescence in DN.

### 3.4. Identification of Potential miR-378i-Skp2 Interaction in Diabetic PTECs

A large body of evidence shows that miRNAs regulate cellular senescence [19,20]. Based on our findings that Skp2 may be involved in PTEC senescence, we investigated whether loss of Skp2 is via miRNAs in the process of PTEC senescence. The heat map of potentially involved miRNAs in normal and diabetic PTECs is shown in Figure 4A. Of 67 miRNAs differentially expressed, three miRNAs had up-expression and 64 miRNAs had down-expression. Of the three up-expressed miRNAs, the pathophysiological function of miR-378i was correlated to cell cycle and G1/S check point regulation, as the top TOX function, as determined by core analysis of IPA database (Figure 4B). We examined miR-378i expression in PTECs using RT-PCR. miR-378i levels were higher in PTECs of a DM patient than those of a normal individual (Figure 4C), and HG induced elevated miR-378i levels in human PTECs after treatment for three days (Figure 4D).

We further utilized miRNA target filter of IPA to search the downstream target of miR-378i, which contributes to cell senescence in PTECs, and found that miR-378i targeted Skp2 and regulated Skp2 expression (Table 3). Next, we used TargetScan (7.1 version), which is a web server that predicts the biological targets of miRNAs to search for the presence of sites that matched the seed region of miRNA. The results showed that miR-378i targeted the position 1889–1895 of Skp2 3’ UTR (Figure 4E), with the context score percentile of 91, which meant that the miR-378i-Skp2 link was highly practicable. Transfection of miR-378i mimic decreased the level of Skp2 in HEK2 cells (Figure 4F). In short, miR-378i modulated Skp2 expression and miR-378i-Skp2 interaction may participate in cell senescence of human PTECs.

### 3.5. Urinary miR-378i Levels are Positively Correlated with Urinary SASP Levels and Albuminuria in Mice and Type 2 DM Patients

Senescent cells have been known to release SASP cytokines, such as IL-6, IL-8, and MCP-1, which can affect signal transduction in neighboring cells and surrounding tissue [8,9]. Since our results demonstrated the interaction between miR-378i and Skp2 in cell senescence of human PTECs, we further investigated the association of miR-378i with SASP and albuminuria in diabetic animal and human studies. Urinary miR-378i levels were measured in db/m mice (*n* = 6) and db/db mice (*n* = 6). The metabolic parameters of mice are shown in Table 4. The level of urinary miR-378i was higher in db/db mice than in db/m mice (Figure 5A). Similarly, SASP cytokines, such as IL-6 and MCP-1, were higher in the urine of db/db mice than db/m mice (Figure 5B,C). High urinary miR-378i/Cr levels were significantly correlated with urinary IL-6/Cr and MCP-1/Cr levels (Figure 5D,E). Urinary miR-378i/Cr, IL-6/Cr, and MCP-1/Cr levels were positively correlated with log-formed urinary ACR in mice (Figure 5F–H).

Forty-five normal individuals and 107 type 2 DM patients were enrolled in our study (Table 5). There was no significant difference of age and sex distribution between the two groups. Thirty-three percentage of type 2 diabetic patients used ACEI/ARB. We found that urinary miR-378i levels were elevated in the type 2 DM patients as compared to normal individuals (Figure 6A). Type 2 DM patients also had higher urinary IL-6/Cr, IL-8/Cr, and MCP-1/Cr levels than normal individuals (Figure 6B–D). A positive association of urinary miR-378i/Cr levels with urinary IL-6/Cr, IL-8/Cr, and MCP-1/Cr levels is presented in Figure 6E–G. Urinary miR-378i/Cr, IL-6/Cr, IL-8/Cr, and MCP-1/Cr levels were positively correlated with urinary ACR (Figure 6H–K). Human participants with macroalbuminuria had higher urinary miR-378i levels, IL-6, IL-8, and MCP-1, than those with normoalbuminuria (Figure 6L–O). After adjusting ACEI/ARB, albuminuria was still significantly and positively associated with urinary miR-378i, IL-6, IL-8, and MCP-1 (Table 6). Besides, urinary miR-378i/Cr, IL-6/Cr, IL-8/Cr, and MCP-1/Cr levels were negatively correlated with eGFR (Figure 6P–S). Consistent with the results of the in vitro study, miR-378i participates in the mechanism of cell senescence-mediating kidney injury, and urinary miR-378i has the potential for predicting kidney dysfunction.

## 4. Discussion

DN develops in one-third of all DM patients, and it accounts as the major cause of ESRD worldwide [28]. Understanding DN mechanisms more clearly, together with early detection of DN onset and progression, greatly assists in avoiding the consequent complications of DN. This cross-disciplinary study demonstrates that HG induces PTEC senescence by enhancing miR-378i expression and reducing Skp2 expression. Furthermore, elevated urinary miR-378i levels are found in diabetic db/db mice and DM patients, and they are positively correlated with SASP, including IL-6, IL-8, and MCP-1, in the urine. The positive association of renal dysfunction with urinary miR-378i is seen in DM patients. This is the first study to provide new perceptions of the unique regulation of miR-378i-Skp2, illustrating the role of PTEC senescence in DN development (Figure 7).

Cell senescence halts PTEC turnover and repair processes and causes progressive inflammation and renal injury [3]. SASP is a possession of senescent cells that coexists with genomic damage and epigenetic abnormality [29]. SASP has played a principal role in tissue microenvironments of renal pathophysiology and it provides biological signal transduction in neighboring cells [30]. SASP promotes the spread of senescence to nearby cells [11] and it further boosts chronic and low-grade inflammation that interrupts homeostasis and results in the development of diseases [29]. Cell senescence participates in pathophysiologic mechanisms of DN [13] and our study shows that HG induces senescence in PTECs.

Based on the bioinformatics results, we found that Skp2 is involved in cell senescence in PTECs in DN. Skp2, a well-characterized F-box protein, is a crucial component of ubiquitin protein ligase complex, called SCF (SKP1-cullin-F-box) [31], which targets phosphorylated cyclin-dependent kinase inhibitor (p27(Kip1) and p21(Cip1)) for ubiquitin modification and subsequent proteasome degradation, thereby facilitating the activation of cdk2-cyclin E complex for progression into the S-phase during the G1/S transition of the cell cycle [32]. Overexpression of Skp2 leads to cell proliferation and tumorigenesis, and it is associated with the down-regulation of p27 [33]. Skp2 deficiency restricts cell development through the up-regulation of p21, p27, and ATF4 [34]. Down-regulation of Skp2 inhibited mesangial cell proliferation [35,36], and Skp2 accumulation causes podocyte injury [37]. However, the role of Skp2 in PTECs is unknown in DN. In the present study, we have demonstrated that HG induces PTEC senescence by reducing Skp2 expression in PTECs, which stops cell development, leading to cell senescence [34]. Therefore, we have concluded that Skp2 contributes to PTEC senescence, further resulting in DN progression.

miRNAs found in biological fluids have been suggested to act in cell-cell communication or as endocrine genetic signals during physiological or pathophysiological processes [38]. The stability of miRNAs in the circulation and in body fluids, their tissue, and disease specificity, and their easy, reliable quantification methods make them feasible as potential biomarkers [39]. miRNAs in the urine have been regarded as potential biomarkers of renal injury [40]. Combination of NGS and IPA analysis confirms that miR-378i has potential activity in the regulation of cell cycle, and Skp2 is the downstream target of miR378i. miR378i has been reported to suppress cell proliferation in hepatocellular carcinoma and colon cancer [41,42]. Our results demonstrate that HG increases the expression of miR-378i in PTECs, and elevated miR-378i is also found in PTECs of type 2 DM patients, supporting the hypothesis that miR-378i contributes to the progression of DN. Computational analysis using IPA shows that miR-378 has a regulatory activity on the cell cycle, and miR-378i mimics decreases Skp2 expression, further corroborating the finding that miR-378i is involved in the cell cycle arrest of senescent PTECs. Moreover, the significantly positive association of urinary miR-378i with urinary SASP, including IL-6, IL-8, and MCP-1, was found in both db/db mice and type 2 DM patients. miR-378i plays a key role in the regulation of PTEC senescence in DN. These results, based on experimental cell studies, animal models, and clinical patients, strongly suggest that miR-378i is one of the critical effectors in PTEC aging enhancing the development of DN in type 2 DM patients.

To examine the predictive activity of miRNAs in renal injury due to DM, we measured urinary miR-378i levels in mice and humans. Our findings show that both diabetic mice and type 2 DM patients have higher miR-378i levels in the urine, and elevated urinary miR-378i is associated with lower eGFR and high albuminuria, a critical renal injury marker. Albuminuria traditionally reflects the glomerular injury. Interestingly, a recent study that was conducted by Wagner et al. provided a new insight that the proximal tubule itself can directly regulate plasma albumin levels through compensating for excessive albumin loss by increasing reclamation [43]. Albuminuria presents as not only glomerular injury, but also the proximal tubular damage [44,45], and it can be considered as the marker of proximal tubule. However, it is difficult to differentiate the origin of albuminuria. Detecting proximal tubular markers (megalin, cubulin) or podocyte indicator (podocin) will help in identifying the source of albuminuria. This is the limitation of this study. Nevertheless, miR-378i is still a potential biomarker that can indicate renal injury in DN and it provides a mechanism for recognizing the early signs of onset or poor progression of DN in clinical practice.

## 5. Conclusions

Our study demonstrated that the interaction between miR-378i and Skp2 induces PTEC senescence under the HG condition. miR-378i has predictive activity for renal injury in DN. This study analyzes the new pathophysiological mechanism of cell senescence of DN, and it provides future applications in the prediction of DN progression.

## Figures and Tables

**Figure 1 jcm-07-00468-f001:**
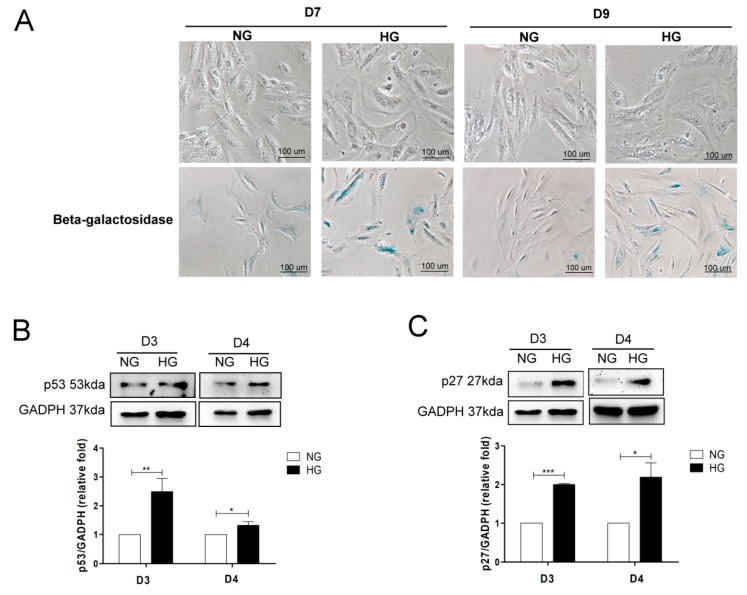
High glucose (HG) induces cell senescence in proximal tubular epithelial cells (PTECs). (**A**) The effect of HG on morphological changes and senescence-associated β-galactosidase (SAβ-Gal) staining of human PTECs. PTECs were incubated under normal glucose (NG, 6.2 mM) and HG (30 mM) conditions for seven days and nine days. Cell senescence was assessed using SAβ-Gal staining. HG increased p53 (**B**) (D3: *n* = 6, D4: *n* = 4) and p27 (**C**) (D3: *n* = 3, D4: *n* = 3), and decreased cyclin E1 (**D**) (D3: *n* = 3, D4: *n* = 4) and cyclin E2 (**E**) (D3: *n* = 3, D4: *n* = 4) protein expression in human PTECs after three and four days of treatment. Protein levels were assessed by western blot. The expression of p53 in the proximal tubule of kidneys of mice (**F**) and humans (**G**). The kidney sections of C57BL/6 mice, non-diabetic db/m mice, and diabetic db/db mice, and human donors (upper tract urothelial carcinoma, UTUC with normal kidney function and normal glomerulus and proximal tubule) and patients with diabetic nephropathy (DN) were stained with p53 (brown). The images quantification was performed using the IHC Profiler Plugin of ImageJ Software. The bar graph represents the mean ± S.E.M. ^*^
*p* < 0.05, ^**^
*p* < 0.01, ^***^
*p* < 0.001 by Student’s *t* test.

**Figure 2 jcm-07-00468-f002:**
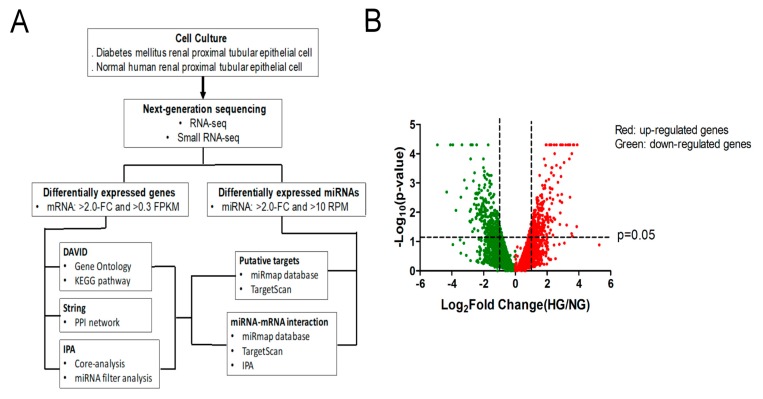
Identification of potential genes associated with cell senescence in PTECs in DN. (**A**) Flowchart of identification of potential genes associated with cell senescence in PTECs. (**B**) Display of differential expression patterns of normal and diabetic PTECs from deep RNA sequencing by volcano plot. (**C**) The Kyoto Encyclopedia of Genes and Genomes (KEGG) pathway enrichment analysis of differentially expressed genes in DAVID database. The 612 differentially expressed genes in diabetic PTECs were uploaded into DAVID database for enrichment analysis. The top seven KEGG pathway analysis results of these dysregulated genes in diabetic PTECs are displayed in a pie chart. The pie chart indicates the-Log10 (false discovery rate, FDR) of each KEGG term, and the numbers that are shown at the outside of each pie segment indicates the number of genes involved in each term. (**D**) The protein-protein interaction network analysis of 14 genes associated cell cycle of KEGG pathway using STRING database. S-phase kinase protein 2 (Skp2) correlated with cell cycle markers, such as cyclin D2 (CCND2), cyclin B1 (CCNB1), cyclin-dependent kinase inhibitor 1C (CDKN1C), and CDKN2C. (**E**) The potential network of Skp2 mediating cell cycle in Core analysis of Ingenuity Pathway Analysis (IPA) software. Skp2 correlated with cyclins.

**Figure 3 jcm-07-00468-f003:**
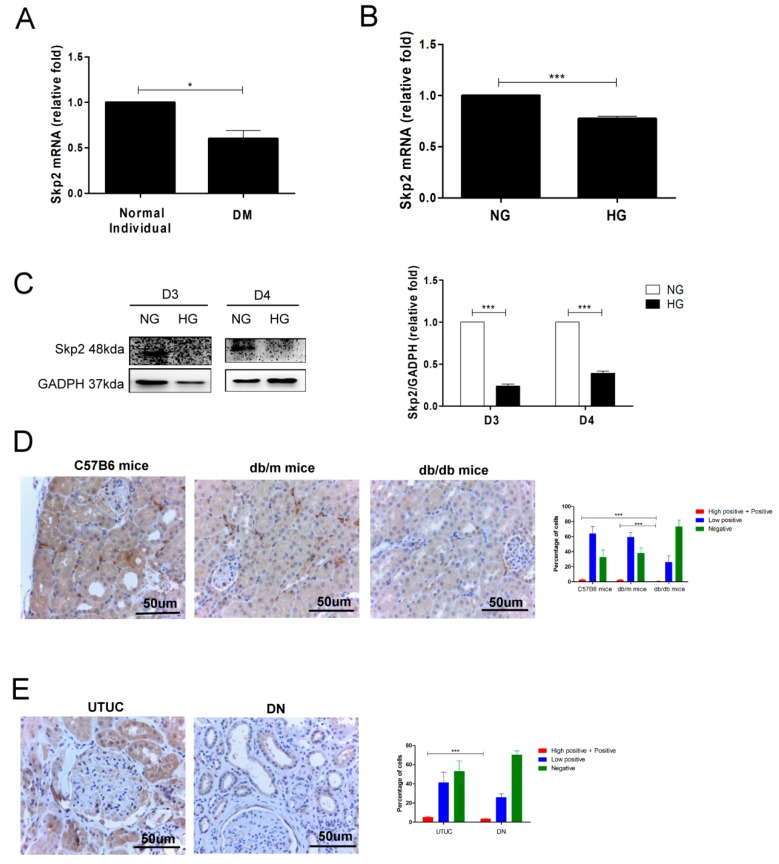
Decreased Skp2 expression is associated with cell senescence of PTECs in DN. (**A**) Decreased Skp2 mRNA expression was found in PTECs from a type 2 diabetes mellitus (DM) patient (*n* = 3). (**B**) HG decreased Skp2 mRNA expression in human PTECs after three days of treatment (*n* = 4). Skp2 mRNA levels were assessed by quantitative real-time polymerase chain reaction (PCR). (**C**) HG suppressed Skp2 protein expression in human PTECs after three and four days of treatment (D3: *n* = 3, D4: *n* = 3). Skp2 protein levels were assessed by western blot. The expression of Skp2 in the proximal tubule of kidneys of mice (**D**) and humans (**E**). The kidney sections of C57BL/6 mice, non-diabetic db/m mice, and diabetic db/db mice, and human donors (UTUC with normal kidney function and normal glomerulus and proximal tubule) and patients with DN were stained with Skp2 (brown). The images quantification was performed using the IHC Profiler Plugin of ImageJ Software. The bar graph represents the mean ± S.E.M. ^*^
*p* < 0.05, ^***^
*p* < 0.001 by Student’s *t* test.

**Figure 4 jcm-07-00468-f004:**
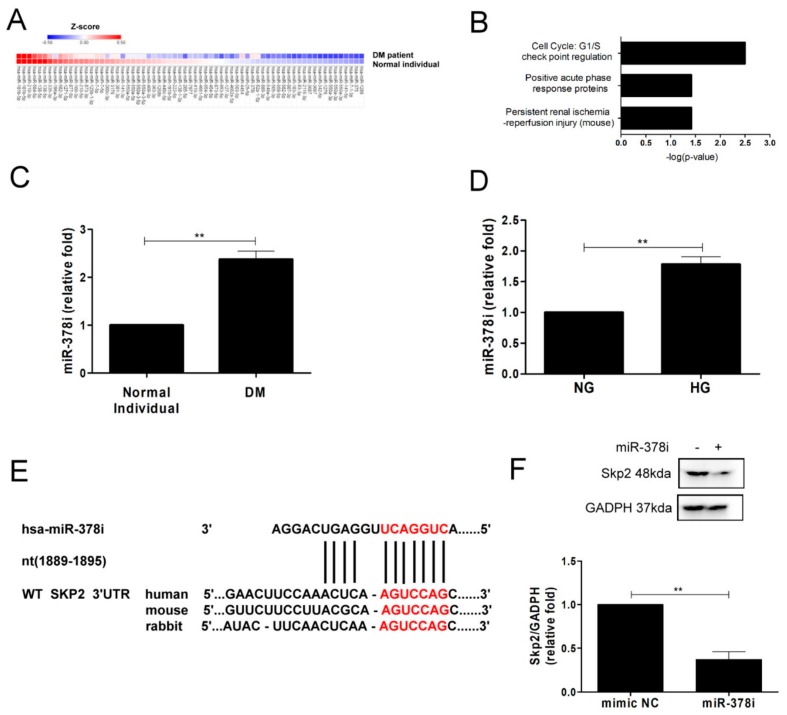
Identification of miR-378i-Skp2 interaction in cell senescence in diabetic PTECs. (**A**) The heat map revealed differentially expressed miRNAs from normal and diabetic PTECs with Z-score values. (**B**) The regulation of miR-378i on cell cycle, as predicted by IPA. (**C**) Increased miR-378i levels were found in PTECs from a type 2 DM patient (*n* = 3). (**D**) HG increased miR-378i levels in human PTECs after three days of treatment (*n* = 3). miR-378i levels were assessed by quantitative real-time PCR. (**E**) A schematic representation of sequence alignment of miR-378i and Skp2 mRNA 3’UTR. (**F**) miR-378i mimic suppressed Skp2 expression in HEK 293 cells. Cells were transfected with either control mimic or miR-378i mimic (100 nM) using DharmaFECT No. 1 Transfection Reagent. After 72 h transfection, western blot was utilized to measure Skp2 protein expression. The bar graph represents the mean ± S.E.M. ^**^
*p* < 0.01 by Student’s *t* test.

**Figure 5 jcm-07-00468-f005:**
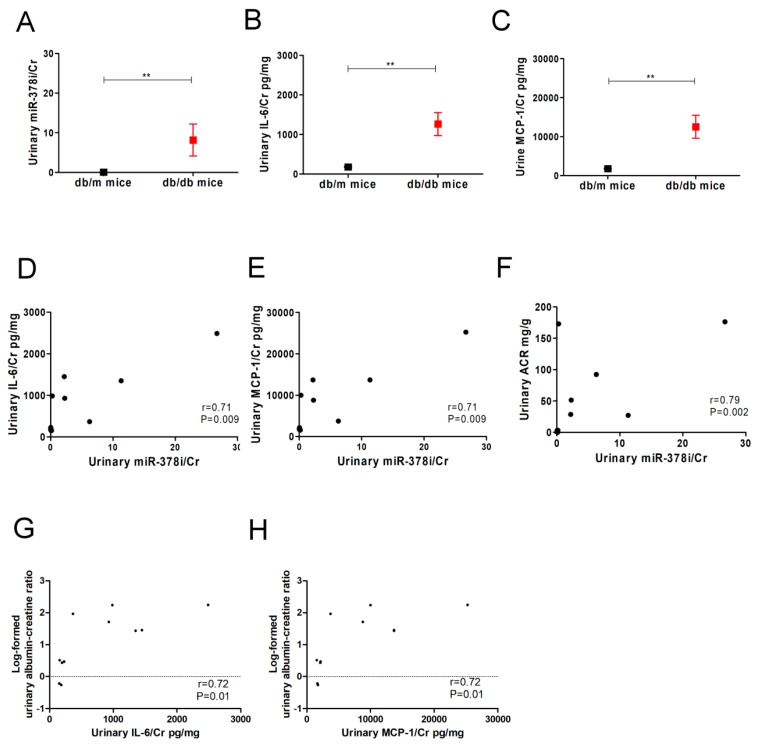
Urinary miR-378i is positively correlated with senescence-associated secretory phenotype (SASP) and renal dysfunction in mice. (**A**) Urinary miR-378i levels were higher in db/db mice (*n* = 6) compared to db/m mice (*n* = 6). Urinary SASP levels, including interleukin-6 (IL-6) (**B**) and monocyte chemoattractant protein-1 (MCP-1) (**C**) were higher in db/db mice than db/m mice. Urinary-378i was positively correlated with IL-6 (**D**) and MCP-1 (**E**) respectively. High urinary miR-378i (**F**), IL-6 (**G**), and MCP-1 (**H**) were significantly correlated with log-formed urinary albumin-creatinine ratio (ACR) in mice. Exosomal miR-378i in the urine of mice was isolated, then assessed by quantitative real-time PCR normalized to a reference control. Magnetic Luminex^®^ Assay was used to assess IL-6 and MCP-1 in the urine of mice. Urine albumin was measured using immunoturbidimetric assay, and urine creatinine was determined by an enzymatic method. The bar graph represents the mean ± S.E.M. ^**^
*p* < 0.01 by Student *t* test, and *p*-value of correlation was analyzed by Spearman analysis. The unit of Cr as mg/dL. The unit of IL-6 and MCP-1 as pg/mL. Cr: creatinine.

**Figure 6 jcm-07-00468-f006:**
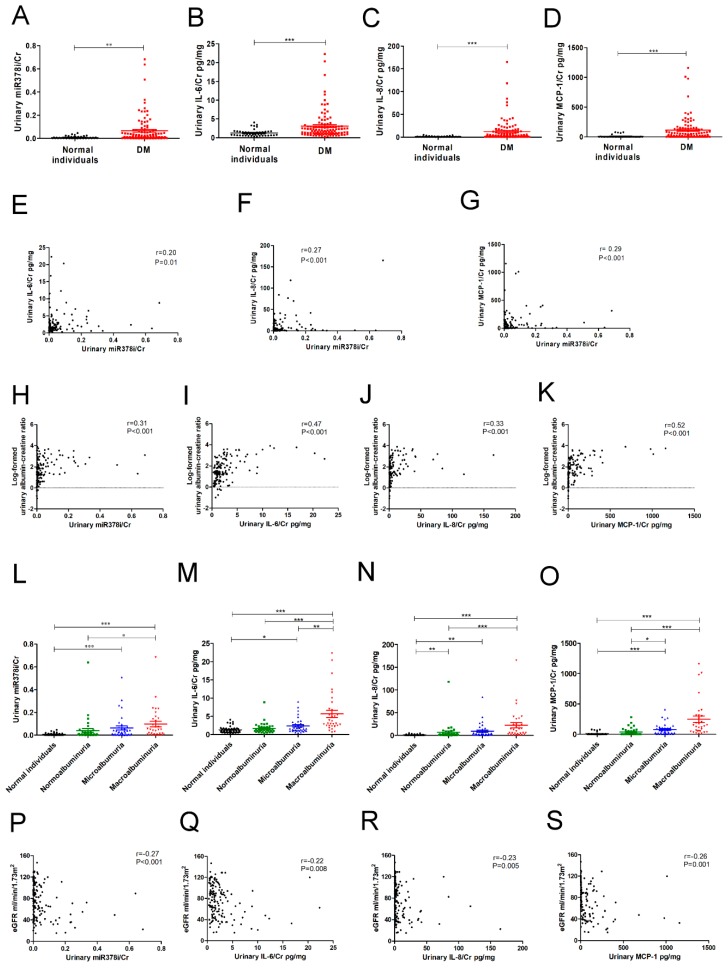
Urinary miR-378i is positively correlated with SASP and renal dysfunction in type 2 DM patients. (**A**) Urinary miR-378i levels were higher in type 2 DM patients (*n* = 107) compared to healthy individuals (*n* = 45). Urinary SASP levels, including interleukin-6 (IL-6) (**B**), IL-8 (**C**), and monocyte chemoattractant protein-1 (MCP-1) (**D**), were higher in type 2 DM patients than in healthy individuals. Urinary miR-378i levels were positively correlated with SASP, including IL-6 (**E**), IL-8 (**F**), and MCP-1 (**G**), in the urine of humans. Urinary miR-378i levels (**H**), IL-6 (**I**), IL-8 (**J**), and MCP-1 (**K**) were positively correlated with log-formed urinary albumin/creatinine ratio (ACR) in human participants. Human participants with macroalbuminuria had higher urinary miR-378i levels (**L**), IL-6 (**M**), IL-8 (**N**), and MCP-1 (**O**), than those with normoalbuminuria. Urinary miR-378i levels (**P**), IL-6 (**Q**), IL-8 (**R**), and MCP-1 (**S**) were negatively correlated with estimated glomerular filtration rate (eGFR) in human participants. Exosomal miR-378i in the urine of humans was isolated, then assessed by quantitative real-time PCR normalized to a reference control. Magnetic Luminex^®^ Assay was used to assess urinary IL-6, IL-8, and MCP-1. Urine albumin was measured using immunoturbidimetric assay and urine creatinine was determined by an enzymatic method. Normoalbuminuria is defined as urinary ACR <30 mg/g; microalbuminuria is defined as urinary ACR ≥30 mg/g and <300 mg/g; macroalbuminuria is defined as urinary ACR ≥300 mg/g. The bar graph represents the mean ± S.E.M. ^*^
*p* < 0.05, ^**^
*p* < 0.01, ^***^
*p* < 0.001 by Student *t* test or ANOVA, followed by the post hoc test adjusted with a Tukey correction, and *p*-value of correlation was analyzed by Spearman analysis. The unit of Cr as mg/dL. The unit of IL-6 and MCP-1 as pg/mL. Cr: creatinine. The unit of eGFR as mL/min/1.73m^2^.

**Figure 7 jcm-07-00468-f007:**
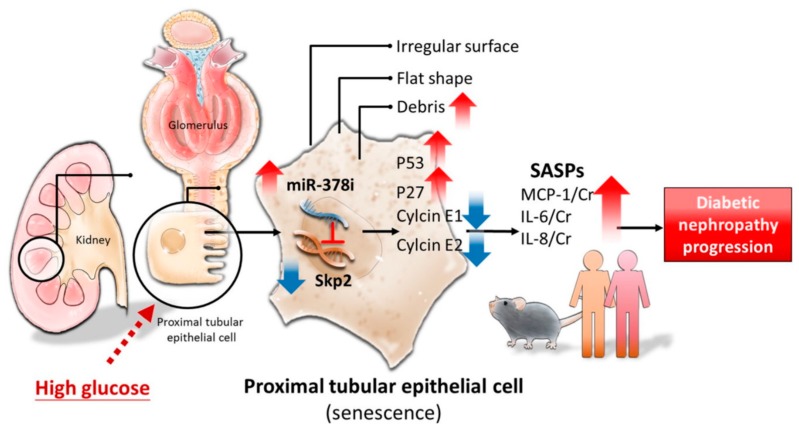
Illustration of the interaction between miR-378i and Skp2 inducing cell senescence in PTECs in DN.

**Table 1 jcm-07-00468-t001:** Target sequence of materials utilized in the study.

Mouse Glyceraldehyde-3-phosphate dehydrogenase (GAPDH)	Forward AACTTTGGCATTGTGTGGAAGG
	Reverse ACACATTGGGGGTAGGAACA
Homo Glyceraldehyde-3-phosphate dehydrogenase (GAPDH)	Forward GAGTCAACGGATTTGGTCGT
	Reverse TTGATTTTGGAGGGATCTCG
hsa-miR-378i	5′d ACTGGACTAGGAGTCAGAAGG 3′
mmu-miR-378a-3p	5′d ACTGGACTTGGAGTCAGAAGG 3′
miRIDIAN microRNA Human hsa-miR378i-Mimic	ACUGGACUAGGAGUCAGAAGG
Homo SKP2	Forward AATTTGCCCTGCAGACTTTG
	Reverse CTGGAGATTCTTTCTGTAGCCG

**Table 2 jcm-07-00468-t002:** The networks associated with genes differentially expressed in diabetic PTECs in IPA database.

	Top Diseases and Functions	Score	Focus Molecules	Molecules in Network
1	Cell Cycle, Cell Morphology, Cellular Movement	46	27	ADAM11, AKR1B10, AKR1D1, caspase, CDK15, cytochrome C, FAM83G, H1F0, HSPA13, KLF4, LAMP3, MMD, MSR1, NMDA Receptor, NT5E, NUPR1, PARP, Pkc(s), PRKCE, PXK, RAB32, RAB39B, RCN3, SLC30A2, SLC47A1, SLC6A9, Sos, STAT5a/b, STC2, TICRR, Tnf (family), TRIM29, ULBP1, XAF1, XBP1
2	Cellular Development, Cellular Growth and Proliferation, Connective Tissue Development and Function	37	23	ACVR1, ANK1, AURK, Cbp/p300, CCNF, CEBPA, CKB, Ctbp, DKK3, FAM49A, GSTM1, H2AFZ, Hdac, HIST1H2BA, HISTONE, histone deacetylase, Histone h3, Histone h4, HMG CoA synthase, Hsp90, INHBE, JMY, KLF15, MEF2A, MEF2C, NDRG2, NPM2, NUAK2, P38 MAPK, PPARG, PSTPIP2, RNA polymerase II, SESN2, TBX15, TFAP2B
3	Cellular Compromise, Cellular Function Maintenance, Immunological Disease	18	14	Alp, AMPK, ASNS, ATF3, CTH, DDIT3, DPEP1, ERK, ERN1, GADD45, GDF15, GNRH, GOT, GOT1, Growth hormone, HDL, HDL-cholesterol, hemoglobin, IFN Beta, IgG, IgG2a, Igm, IL12 (complex), Ldh (complex), LDL, LIPG, NADPH oxidase, NFIL3, PCK2, PI3K (family), PPP1R3G, PRKAA, Pro-inflammatory Cytokine, SAA2, TRIB3
4	Cellular Development, Cellular Growth and Proliferation, Organ Development	15	12	Akt, calpain, Cdc2, Cdk, CDO1, Collagen type IV, Collagen(s), Cyclin A, Cyclin D, Cyclin E, E2f, ELN, FBLN1, FGF2, Fibrin, Fibrinogen, GABP, gelatinase, HEY1, Laminin (complex), MLPH, ORC1, Pdgf (complex), PDGF BB, PDGF-AA, PLAT, PRKG2, Ptk, Rb, SERPINF, SERPINF2, SKP2, trypsin, Wnt, ZFP57
5	Cellular Growth and Proliferation, Hair and Skin Development and Function, Cancer	13	11	AGT, Ap1, BNC1, Calmodulin, Cg, EREG, FSH, G protein alphai, GLI2, Gpcr, Gpd, ID4, Insulin, Lh, LZTS1, Mapk, Mek, Mmp, p70 S6k, Pka, Pka catalytic subunit, PLA1A, PLC, PPEF2, Ras, Sfk, SFRP1, Shc, SLC4A4, Smad, Smad2/3, SSTR5, Tgf beta, Vegf, voltage-gated calcium channel

**Table 3 jcm-07-00468-t003:** Potential microRNA–mRNA interactions identified in diabetic PTECs.

miRNA	Precursor	Log2 Ratio	Fold Change	DM Seq (norm)	Non-DM Seq (norm)	DM Read Count	Non-DM Read Count	Target Gene	Fold Change
hsa-miR-378i	hsa-mir-378i	1.02	2.02	9.61	4.76	111	53	Skp2	−1.504
hsa-miR-92a-1-5p	hsa-mir-92a-1	1.05	2.07	9.09	4.4	105	49	ORC1	−1.666
hsa-miR-4454	hsa-mir-4454	1.07	2.11	11.35	5.39	131	60		

PTEC: Proximal tubular epithelial cell; DM: Diabetes Mellitus.

**Table 4 jcm-07-00468-t004:** The metabolic parameters of mice.

	db/m mice *n* = 6	db/db mice *n* = 6	*p*-Value
Post-meal blood glucose, mg/dL	204.6 ± 65.9	521.8 ± 44.7	<0.001
Blood urea nitrogen, mg/dL	12.9 ± 1.6	42.7 ± 21.5	0.02
Serum creatinine, mg/dL;	0.0 ± 0.1	0.1 ± 0.1	0.02
Urinary albumin/creatinine ratio, mg/g	26.9 (3.0–30.8)	717.9 (282.1–1737.7)	<0.001

**Table 5 jcm-07-00468-t005:** The characteristics and metabolic parameters of human participants.

	Normal Individuals *n* = 45	Type 2 Diabetes *n* = 107	*p*-Value
Age, years	60.4 ± 6.6	63.3 ± 11.0	0.05
Sex (male), %	53.3	54.2	0.92
Fasting blood glucose, mg/dL	107.9 ± 33.4	146.2 ± 51.9	<0.001
Blood urea nitrogen, mg/dL	15.2 ± 3.6	20.2 ± 9.3	<0.001
Serum creatinine, mg/dL	0.8 ± 0.2	1.2 ± 0.7	<0.001
Estimated glomerular filtration rate, mL/min/1.73m^2^	97.5 ± 19.4	66.8 ± 29.7	<0.001
Urine albumin/creatinine ratio, mg/g	2.8 (1.4–4.7)	77.0 (22.9–596.5)	<0.001
ACEI/ARB usage, %		33.8	

Abbreviation: ACEI, angiotensin-converting enzyme; ARB, angiotensin II receptor blocker.

**Table 6 jcm-07-00468-t006:** Determinants of albuminuria using multivariate linear analysis in human participants.

Urinary Parameters	Unstandardized Coefficient β (95% CI)	*p*-Value
miR378i/Cr	2.616 (0.863–4.369)	0.004
IL-6/Cr, pg/mg	0.097 (0.058–0.135)	<0.001
IL-8/Cr, pg/mg	0.015 (0.007–0.023)	<0.001
MCP-1, pg/mg	0.002 (0.001–0.003)	0.003

Values expressed as unstandardized coefficient β and 95% confidence interval (CI). Adjust for age, sex and angiotensin-converting enzyme/angiotensin II receptor blocker usage.

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
