# Peer review of "The Interaction of miR-378i-Skp2 Regulates Cell Senescence in Diabetic Nephropathy"

_jcm, 2018, doi:10.3390/jcm7120468_

Reviewer 1 Report

In this manuscript, the authors have examined the interactions of miR-378i-Skp2 in the regulation of cell senescence in diabetic nephropathy. They observed and state that miR-378i is the marker of renal injury under diabetic state and HG ambience. The study plan is good and innovative. After incorporating changes as mentioned below manuscript can be accepted. Here, some points that authors should consider.

Minor Revisions:

1.    In the introduction section, some lines and references must be incorporated which would suggest continuity between DN and microRNA (after paragraph 3-line number 71 and 72). The concept of microRNA starts at random. The link between DN and microRNA is well established, but the author should make it more cohesive.

2.    In the section of Material and Method was 0.5% serum was used or was it 5%. Whereas for HEK293T cells 10% serum was used. We understand specific cell lines have different requirement of serum, author should make it more vivid (Primary cells need more serum than HEK293T).  The usage of difference in serum can cause significant variations in results.

3.    In Results section 3.3. line number 293, the author may consider replacing phrase western blot analysis found that with western blot analysis showed.

4.    Results discussed in section 3.4 and what is written in figure legends do not match. According to my understanding, miR-378i expression increased in DM and HG. But in legends, it is written other way. Please correct it accordingly

Author Response

Reviewer 1

Comments and Suggestions for Authors

In this manuscript, the authors have examined the interactions of miR-378i-Skp2 in the regulation of cell senescence in diabetic nephropathy. They observed and state that miR-378i is the marker of renal injury under diabetic state and HG ambience. The study plan is good and innovative. After incorporating changes as mentioned below manuscript can be accepted. Here, some points that authors should consider.

Minor Revisions:

1.          In the introduction section, some lines and references must be incorporated which would suggest continuity between DN and microRNA (after paragraph 3-line number 71 and 72). The concept of microRNA starts at random. The link between DN and microRNA is well established, but the author should make it more cohesive.

Ans: Thank you for your suggestion. We had revised the description at the paragraph 3 and 4 of introduction part in page 2 to make it more cohesive.

2.          In the section of Material and Method was 0.5% serum was used or was it 5%. Whereas for HEK293T cells 10% serum was used. We understand specific cell lines have different requirement of serum, author should make it more vivid (Primary cells need more serum than HEK293T). The usage of difference in serum can cause significant variations in results.

Ans: Thank you for your opinion. Human proximal tubular epithelial cells (PTECs) were cultured in Renal Epithelial Cell Basal Medium (ATCC PCS400030TM) plus 0.5% Fetal Bovine Serum (FBS) according to manufacturer’s suggestion. We did not add FBS in order to avoid the effect of excess growth factors that may disturb the results of experiments. Besides, we used HEK293 cells to identify Skp2 as the downstream target of miR378i. The viability of HEK293 cells is low under culture medium with low percentage of FBS (J Biosci Bioeng. 2014 Apr;117(4):471-7). Therefore, we culture HEK293 cells in DMEM containing 10% FBS according to manufacturer’s suggestion.

3.          In Results section 3.3. line number 293, the author may consider replacing phrase western blot analysis found that with western blot analysis showed.

Ans: Thank you for your suggestion. We added “showed” in this sentence to replace “found”.

4.          Results discussed in section 3.4 and what is written in figure legends do not match. According to my understanding, miR-378i expression increased in DM and HG. But in legends, it is written other way. Please correct it accordingly 

         Ans: Thank you for your correction. We had revised legend 4 correctly.

Reviewer 2 Report

The authors compared small RNAs in commercial primary proximal tubular epithelial cells from a type 2 diabetic donor and a normal donor (Lonza Walkersville Inc) using deep sequencing of RNA(Illumina Inc.). They found miR-378i and its target mRNA S-phase kinase protein (Skp2) contribute to high glucose-induced senescence in diabetic renal proximal tubular cells. The authors showed elevated urinary miR-378i and decreased renal miR-378i-targetted Skp2 in db/db mice and type 2 diabetic patients. In addition, they found the positive correlations between urinary miR-378i and urinary SASP cytokines such as IL-6, IL-8 or MCP-1 levels in mice and diabetic patients

Major comments.

1.  There is no data regarding the metabolic parameters of mice and diabetic patients in the manuscript. The authors should present the information of mice and patients and the definition of diabetes and diabetic nephropathy. Then, the potential confounders such as medication should be taken into account.

2.  The authors represented the immunohistochemical study for p53 and Skp2 in Fig. 1 F, G and Fig. D, E. The intensity and extent of positive immunoreactivity for these immunohistochemical analysis should be presented.

3.  All nonnormally distributed variables such as urinary albumin should be logarithmic transformed for statistical analysis.

4.  The performed times for in vitro experiments should be stated.

5.  The authors showed higher expression of miR-378i and lower expression of Skp2 in diabetic proximal tubular epithelial cells (PTECs) in cultured compared to those in normal PTECs. In addition, they demonstrated that elevated urinary miR-378i, which might be derived from damaged PTECs, is associated with high albuminuria. However, albuminuria usually reflects the damage of diabetic glomeruli. The authors should clarify the origin of albuminuria by analyzing the glomerular and tubular injury in diabetic patients.

6.  The authors showed the effects of miR378 mimic on Skp2 expression in Figure 4F, and high glucose decreases Skp2 expression and induces cell senescense in PTECs. Taken together, the authors interpret that miR-378i-Skp2 contributed to high glucose-induced PTEC senescence. However, a direct functional correlation between Skp2 and cell senescence is lacking in the present manuscript. More direct experimental evidence would be desirable. The authors should investigate the effects of miR-378i inhibitor on high glucose-inhibited Skp2 mRNA and overexpressing the expression of Skp2 on high glucose-induced senescence expression.

7.  In Results in line 383 of Page 384, the authors state that urinary miR-378i has the potential for predicting kidney dysfunction, showing the correlations between urinary miR-378i and urinary albumin, IL-6, IL-8 and MCP-1. To refer the renal function, the authors should present the data about GFR or surrogate for GFR.

Minor comments.

1.  The authors stated that urinary miR-378i levels are positively correlated with SASP, IL-6, IL-8 and MCP-1 in the urine. However, IL-8 data in mice is missing.

2.  Figure legend for Figure 4 is incorrect. The authors should rewrite it.

Author Response

Reviewer 2

Comments and Suggestions for Authors

The authors compared small RNAs in commercial primary proximal tubular epithelial cells from a type 2 diabetic donor and a normal donor (Lonza Walkersville Inc) using deep sequencing of RNA(Illumina Inc.). They found miR-378i and its target mRNA S-phase kinase protein (Skp2) contribute to high glucose-induced senescence in diabetic renal proximal tubular cells. The authors showed elevated urinary miR-378i and decreased renal miR-378i-targetted Skp2 in db/db mice and type 2 diabetic patients. In addition, they found the positive correlations between urinary miR-378i and urinary SASP cytokines such as IL-6, IL-8 or MCP-1 levels in mice and diabetic patients.

Major comments.

1.      There is no data regarding the metabolic parameters of mice and diabetic patients in the manuscript. The authors should present the information of mice and patients and the definition of diabetes and diabetic nephropathy. Then, the potential confounders such as medication should be taken into account.

Ans: Thank you for your suggestion. Diabetes was defined as a medical history of diabetes or the use of anti-diabetes agents. Diabetic nephropathy was defined as diabetic patients with urine albumin/creatinine ratio over 30mg/g. We added the metabolic parameters, including BUN, Cr and fasting sugar in mice and human in New Table 4 and 5. We also provided angiotensin-converting enzyme (ACEI)/ angiotensin receptor blocker (ARB) usage in diabetic patients, which could reduce albuminuria. After adjusting ACEI/ARB usage, urinary miR-378i, IL-6, IL-8, and MCP-1 were still significantly correlated with albuminuria in type 2 diabetic patients (New Table 6). We would revise 2.12 and 2.13 method part in page 5 and 3.5 result part in page 15-18.

 New Table 4. The characteristics and metabolic parameters of mice

db/m   mice

N=6

 db/db mice

 N=6

P-value

Post-meal   glucose, mg/dl

204.6±65.9

521.8±44.7

<0.001< span="">

Blood   urea nitrogen, mg/dl

12.9±1.6

42.7±21.5

0.02

Serum   creatinine, mg/dl

0.0±0.1

0.1±0.1

0.02

Urinary   albumin/creatinine ratio, mg/g

717.9(282.1,1737.66)

26.9(3.0,30.8)

<0.001< span="">

New Table 5. The characteristics and metabolic parameters of human participants

Normal individuals

N=45

 Type 2 diabetes

 N=107

P-value

Age, years

60.4±6.6

63.3±11.0

0.05

Sex (male), %

53.3

54.2

0.92

Fasting glucose, mg/dl

107.9±33.4

146.2±51.9

<0.001< span="">

Blood urea nitrogen, mg/dl

15.2±3.6

20.2±9.3

<0.001< span="">

Serum creatinine, mg/dl

0.8±0.2

1.2±0.7

<0.001< span="">

Estimated glomerular filtration   rate, ml/min/1.73m2

97.5±19.4

66.8±29.7

<0.001< span="">

Urinary albumin/creatinine   ratio, mg/g

2.8(1.4,4.7)

77.0(22.9,596.5)

<0.001< span="">

ACEI/ARB usage, %

-

33.8

-

Abbreviation: ACEI, angiotensin-converting enzyme; ARB, angiotensin II receptor blocker 

 New Table 6. Determinants of albuminuria using multivariate linear analysis in human participants

Urinary   parameters

Unstandardized coefficient β   (95% CI)

p

miR378i/Cr

    2.616(0.863, 4.369)

0.004

IL-6/Cr,   pg/mg

0.097(0.058, 0.135)

<0.001< span="">

IL-8/Cr,   pg/mg

        0.015(0.007, 0.023)

< 0.001

MCP-1,   pg/mg

  0.002(0.001,   0.003)

0.003

Values expressed as unstandardized coefficient β and 95% confidence interval (CI).

Adjust for age, sex and angiotensin-converting enzyme/angiotensin II receptor blocker usage

 2.  The authors represented the immunohistochemical study for p53 and Skp2 in Fig. 1 F, G and Fig. D, E. The intensity and extent of positive immunoreactivity for these immunohistochemical analysis should be presented.

Ans: Thank you for your suggestion. The images quantification was performed using the IHC Profiler Plugin of ImageJ Software (https://imagej.nih.gov/ij/) (  (PLoS ONE 2014; 9: e96801). We added these analysis results in revised Figure 1 and 3 respectively.

p53

 Skp2

 3.      All nonnormally distributed variables such as urinary albumin should be logarithmic transformed for statistical analysis.

Ans: Thank you for your suggestion. The association of log-formed urinary albumin/creatinine ratio with urinary miR-378i, IL-6, IL-8, and MCP-1 was consistent. We would revise Figure 5 and 6.

4.  The performed times for in vitro experiments should be stated.

Ans: Thank you for your suggestion. We added the performed times for for in vitro experiments in legend 1, 3 and 4.

Legend 1: HG increased p53 (D3: n=6, D4: n=4, Figure 1B) and p27 (D3: n=3, D4: n=3, Figure 1C), and decreased cyclin E1 (D3: n=3, D4: n=4, Figure 1D) and cyclin E2 (D3: n=3, D4: n=4, Figure 1E) protein expression in human PTECs after 3 and 4 days of treatment.

Legend 3: (Figure 3A) Decreased Skp2 mRNA expression was found in PTECs from a type 2 diabetes mellitus (DM) patient (n=3). (Figure 3B) HG decreased Skp2 mRNA expression in human PTECs after 3 days of treatment (n=4). (Figure 3C) HG suppressed Skp2 protein expression in human PTECs after 3 and 4 days of treatment (D3: n=3, D4: n=3).

Legend 4: (Figure 3C) Increased miR-378i levels were found in PTECs from a type 2 DM patient (n=3). (D) HG increased miR-378i levels in human PTECs after 3 days of treatment (n=3).

5.  The authors showed higher expression of miR-378i and lower expression of Skp2 in diabetic proximal tubular epithelial cells (PTECs) in cultured compared to those in normal PTECs. In addition, they demonstrated that elevated urinary miR-378i, which might be derived from damaged PTECs, is associated with high albuminuria. However, albuminuria usually reflects the damage of diabetic glomeruli. The authors should clarify the origin of albuminuria by analyzing the glomerular and tubular injury in diabetic patients.

Ans: Thank you for your opinion. Albuminuria traditionally reflects the glomerular injury. Interestingly, recent study conducted by Wagner et al. provided a new insight that the proximal tubule itself can directly regulate plasma albumin levels through compensating for excessive albumin loss by increasing reclamation. Albuminuria can present not only glomerular injury but also the proximal tubule damage (J Am Soc Nephrol 2014; 25:443-453; Nat Rev Nephrol 2015; 11: 573-575). Albuminuria can be considered as the marker of proximal tubule. However, it is difficult to differentiate the origin of albuminuria. Megalin and cubulin, as well-known markers of proximal tubule, can uptake albumin. Besides, podocin is considered as podocyte indicator. Therefore, to detect megalin, cubulin, or podocin will help identifying the source of albuminuria. This is the limitation of this study. We would state this limitation in last paragraph of discussion section in page 20.

6.  The authors showed the effects of miR378 mimic on Skp2 expression in Figure; 4F, and high glucose decreases Skp2 expression and induces cell senescense in PTECs. Taken together, the authors interpret that miR-378i-Skp2 contributed to high glucose-induced PTEC senescence. However, a direct functional correlation between Skp2 and cell senescence is lacking in the present manuscript. More direct experimental evidence would be desirable. The authors should investigate the effects of miR-378i inhibitor on high glucose-inhibited Skp2 mRNA and overexpressing the expression of Skp2 on high glucose-induced senescence expression.

Ans: Thank you for your valuable suggestion. However, the period of revision is limited, only 7 days. We need more time to investigate the effects of miR-378i inhibitor on high glucose-inhibited Skp2 mRNA and overexpressing the expression of Skp2 on high glucose-induced senescence expression.

7.  In Results in line 383 of Page 384, the authors state that urinary miR-378i has the potential for predicting kidney dysfunction, showing the correlations between urinary miR-378i and urinary albumin, IL-6, IL-8 and MCP-1. To refer the renal function, the authors should present the data about GFR or surrogate for GFR.

Ans: Thank you for your suggestion. We analyzed the relationship between urinary miR-378i and eGFR in human study, and found a positive association of eGFR with urinary miR-378i, IL-6, IL-8 and MCP-1. Urinary miR-378i has the potential for predicting renal dysfunction in clinical diabetes. We added these results in revised Figure 6 in page 18.

Minor comments.

1.      The authors stated that urinary miR-378i levels are positively correlated with SASP, IL-6, IL-8 and MCP-1 in the urine. However, IL-8 data in mice is missing.

Ans: Thank you for your opinion. Because mice don't express IL-8 (or the receptor  CXCR1). Therefore, we did not assess the level of IL8 in mouse model.

2.  Figure legend for Figure 4 is incorrect. The authors should rewrite it.

Ans: Thank you for your correction. We had revised legend 4 correctly.

Round  2

Reviewer 2 Report

Minot comments.

Table 4. The metabolic parameters of mice.Urinary albumin/creatinine ratio of lean control db/m mice is higher than that of diabetic db/db mice.

Table 4. Post-meal blood glucose.

Table 5. Fasting plasma (or blood) glucose.

Author Response

Table 4. The metabolic parameters of mice.Urinary albumin/creatinine ratio of lean control db/m mice is higher than that of diabetic db/db mice.

Ans: Thank you for your correction. We had revised urinary albumin/creatinine ratio in db/m mice and db/db mice in revised table 4.

Revised Table 4

db/m mice

N=6

 db/db mice

 N=6

P-value

Post-meal blood glucose,   mg/dl

204.6±65.9

521.8±44.7

<0.001< span="">

Blood urea nitrogen, mg/dl

12.9±1.6

42.7±21.5

0.02

Serum   creatinine,   mg/dl

0.0±0.1

0.1±0.1

0.02

Urinary albumin/creatinine   ratio, mg/g

26.9(3.0,30.8)

717.9(282.1,1737.7)

<0.001< span="">

Table 4. Post-meal blood glucose.

Ans: Thank you for your suggestion. We revised post-meal blood glucose in revised table 4.

Table 5. Fasting plasma (or blood) glucose.

Ans: Thank you for your suggestion. We revised fasting blood glucose in revised table 5.

Revised Table 5

Normal individuals

N=45

 Type 2   diabetes

 N=107

P-value

Age, years

60.4±6.6

63.3±11.0

0.05

Sex (male), %

53.3

54.2

0.92

Fasting blood glucose, mg/dl

107.9±33.4

146.2±51.9

<0.001< p="">

Blood urea nitrogen, mg/dl

15.2±3.6

20.2±9.3

<0.001< p="">

Serum creatinine, mg/dl

0.8±0.2

1.2±0.7

<0.001< p="">

Estimated glomerular filtration rate,   ml/min/1.73m2

97.5±19.4

66.8±29.7

<0.001< p="">

Urine albumin/creatinine ratio,   mg/g

2.8(1.4,4.7)

77.0(22.9,596.5)

<0.001< p="">

ACEI/ARB usage, %

-

33.8

-
